# Metabolic Rewiring and Altered Glial Differentiation in an iPSC-Derived Astrocyte Model Derived from a Nonketotic Hyperglycinemia Patient

**DOI:** 10.3390/ijms25052814

**Published:** 2024-02-28

**Authors:** Laura Arribas-Carreira, Margarita Castro, Fernando García, Rosa Navarrete, Irene Bravo-Alonso, Francisco Zafra, Magdalena Ugarte, Eva Richard, Belén Pérez, Pilar Rodríguez-Pombo

**Affiliations:** 1Centro de Biología Molecular Severo Ochoa UAM-CSIC, Instituto de Biología Molecular, Departamento de Biología Molecular, Universidad Autónoma de Madrid, 28049 Madrid, Spain; laura.arribas@cbm.csic.es (L.A.-C.); rnavarrete@cbm.csic.es (R.N.); fzafra@cbm.csic.es (F.Z.); mugarte@cbm.csic.es (M.U.); erichard@cbm.csic.es (E.R.); bperez@cbm.csic.es (B.P.); 2Centro de Diagnóstico de Enfermedades Moleculares (CEDEM), 28049 Madrid, Spain; mcastro@cbm.csic.es (M.C.); fgmunioz@cbm.csic.es (F.G.); ibravo@cbm.csic.es (I.B.-A.); 3Centro de Investigación Biomédica en Red de Enfermedades Raras (CIBERER), ISCIII, 28029 Madrid, Spain; 4Instituto de Investigación Sanitaria Hospital La Paz (IdiPaz), ISCIII, 28029 Madrid, Spain

**Keywords:** human disease model, inherited metabolic disorders, inborn error of metabolism, nonketotic hyperglycinemia, *GLDC*, iPSC, iPSC-derived astrocytes, serine-glycine-one-carbon metabolism

## Abstract

The pathophysiology of nonketotic hyperglycinemia (NKH), a rare neuro-metabolic disorder associated with severe brain malformations and life-threatening neurological manifestations, remains incompletely understood. Therefore, a valid human neural model is essential. We aimed to investigate the impact of *GLDC* gene variants, which cause NKH, on cellular fitness during the differentiation process of human induced pluripotent stem cells (iPSCs) into iPSC-derived astrocytes and to identify sustainable mechanisms capable of overcoming *GLDC* deficiency. We developed the GLDC27-FiPS4F-1 line and performed metabolomic, mRNA abundance, and protein analyses. This study showed that although GLDC27-FiPS4F-1 maintained the parental genetic profile, it underwent a metabolic switch to an altered serine–glycine–one-carbon metabolism with a coordinated cell growth and cell cycle proliferation response. We then differentiated the iPSCs into neural progenitor cells (NPCs) and astrocyte-lineage cells. Our analysis showed that *GLDC*-deficient NPCs had shifted towards a more heterogeneous astrocyte lineage with increased expression of the radial glial markers *GFAP* and *GLAST* and the neuronal markers *MAP2* and *NeuN.* In addition, we detected changes in other genes related to serine and glycine metabolism and transport, all consistent with the need to maintain glycine at physiological levels. These findings improve our understanding of the pathology of nonketotic hyperglycinemia and offer new perspectives for therapeutic options.

## 1. Introduction

The distinctive nature of rare human monogenic diseases, characterized by the limited number of patients available worldwide and restricted access to the most affected tissues, poses challenges in unravelling the molecular, cellular, tissue, and organ-level processes that sustain the disease condition and influence the response to treatment. Reprogramming somatic cells to induce pluripotency has revolutionized the approach to modeling human disease. Induced pluripotent stem cells (iPSCs) can differentiate into cells of the three primary germ layers while retaining the genetic background of their somatic origin. In neurological diseases, iPSCs can be induced to differentiate into most cell types found in the human brain and central nervous system, providing a neural model to study disease phenotypes and develop drugs for human diseases [1,2]. One of the rare neurometabolic diseases that could benefit from this approach is nonketotic hyperglycinemia (NKH), a disorder characterized by severe brain malformations and life-threatening neurological manifestations. Classic NKH (MIM#605899) is an autosomal recessive disorder that results from deficient activity of the multi-enzyme mitochondrial complex the Glycine Cleavage System (GCS) [3]. In adults, the GCS is activated in a limited number of tissues, including the liver, brain, lungs, and kidneys, and mediates the decarboxylation of glycine with the concomitant transfer of a one-carbon group to tetrahydrofolate (THF), resulting in the formation of 5,10-methylene THF. In this way, GCS contributes to maintaining the flux of the serine–glycine–one-carbon metabolic pathway. The GCS complex consists of four components: a pyridoxal-dependent glycine decarboxylase (GCSP), encoded by the *GLDC* gene (MIM*238300) responsible for catalyzing the decarboxylation of glycine with CO_2_ release; an amino-methyltransferase, a tetrahydrofolate-dependent GCST protein encoded by the *AMT* gene (MIM*238310); a NAD^+^-dependent dihydro-lipoamide dehydrogenase; and a small lipoylated H protein (GCSH) (*GCSH*; MIM*238330), capable of interacting with the other subunits. GCSH is responsible for the transferring of lipoic acid to other mitochondrial apo-enzymes, including the E2 subunit of the pyruvate dehydrogenase and the α-ketoglutarate dehydrogenase complexes [4,5]. Mutations in *GLDC* or *AMT* genes can lead to the accumulation of glycine in biological fluids, particularly in cerebrospinal fluid, serving as the biochemical hallmark of NKH. More than 70% of NKH patients carry mutations in the *GLDC* gene. Several models are being considered to predict the severity of NKH based on the relationship between the residual activity of nucleotide variants and disease outcome [6,7].

For years, the severe neurological symptoms observed in NKH patients have been associated with an excess of glycine. Glycine acts on both NMDA receptors in the cortex and glycine receptors in the spinal cord and brain stem neurons [8,9]. However, glycine metabolism is closely linked to maintaining the cellular pools of one-carbon residues necessary for synthesizing amino acids, nucleotides, and phospholipids, as well as remodeling the cell’s epigenetic state. Accumulated evidence has demonstrated that the hyperactivity of *GLDC* plays a critical role in sustaining the proliferation of different types of cancer cells [10] or during the reprogramming of somatic cells to pluripotent iPSCs [11]. 

These two faces of glycine as a neurotransmitter and a donor of one-carbon units were previously proposed for explaining the effect of GCS deficiency in the NKH disease condition [12], and this seems to be supported by data in *Gldc*^-/-^ mice, indicating the involvement of impaired folate 1-carbon metabolism [13] in specific aspects of NKH pathogenesis. Additionally, a strong relationship exists between neural-tube disorders (NTD) and *GLDC* or *AMT* mutations [14].

Glycine abundance is modulated in a tissue-specific manner through biosynthetic reactions, catabolism, and excretion via glycine conjugation. However, there is a gap in knowledge regarding the cellular response to GCS deficiency in neural cells. Our analysis focused on astrocytes, which comprise most of the cell population in the human brain and play a key role in serine–glycine–one-carbon metabolism. We utilized a human induced pluripotent stem cell (iPSC) model [15] derived from fibroblasts of a patient with severe NKH with hypomorphic *GLDC* mutations [16] to investigate the effects of *GLDC* deficiency on cell metabolism, bioenergetics, growth, and cell cycle, from pluripotent iPSCs to differentiated neural cells. 

## 2. Results

### 2.1. The Human iPSC Model Recapitulates GLDC Deficiency

To determine the potential of iPSCs to mimic *GLDC* deficiency, we examined the expression of *GLDC*, *AMT*, and *GCSH* in these cells. We evaluated both control and GLDC27-FiPS4F-1 iPSCs [15], obtained by reprogramming dermal fibroblasts using the CytoTune™ iPS Reprogramming kit delivering the four human reprogramming factors *OCT3/4*, *SOX2*, *c-MYC*, and *KLF4* [17]. These cell lines were previously characterized for pluripotency, normal karyotype, and line identity through short tandem repeat analysis. Both were assessed at the same stage of reprogramming. Figure 1A–D present the results of various analyses, including RT-qPCR analysis, steady-state analysis of the GCS proteins (GCSP, GCST, and GCSH) under denaturing or native conditions, and glycine exchange activity in both iPSCs and fibroblasts derived from the donor NKH patient and healthy control. Recapitulating *GLDC* deficiency at the protein level, GCSP protein and GCS activity appeared severely reduced in GLDC27-FiPS4F-1 iPSCs compared to the healthy control. Moreover, the data show that in the human healthy control, the process of somatic reprogramming to pluripotency induced the expression of *GLDC* and the corresponding GCS activity, which was absent in fibroblasts. This suggests a significant role of GCSP protein at the pluripotent stage. Immunofluorescence analysis of GCS proteins in normal and GLDC27-FiPS4F-1 iPSCs (Figure 1E,F) supported and corroborated these results. 

### 2.2. Coordinated Response of Serine–Glycine–One-Carbon Metabolism to GLDC Deficiency in iPSCs

Given the involvement of GCS in the serine–glycine–one-carbon pathways, we assessed the cellular response of iPSCs to *GLDC* deficiency by analyzing primary metabolites and gene expression of key components of the serine–glycine–one-carbon metabolism. Metabolites were quantified in cellular extracts and spent-culture medium of the iPSCs using a targeted metabolomics methodology. Differences in the absorption and excretion of amino acids and short peptides were observed between GLDC27-FiPS4F-1 and control cells. Figure 2A shows a schematic view of amino acid metabolism.

The GLDC27-FiPS4F-1 cells exhibited markedly different behavior regarding glycine concentrations; while the control cells absorbed glycine from the environment, the GLDC27-FiPS4F-1 cells released high levels of glycine from the cell into the environment. In contrast, both cell lines absorbed serine from the culture medium, but the uptake was notably higher in the GLDC27-FiPS4F-1 line (1.5 times that of the control) (Figure 2B). Additionally, although both lines exhibited a pattern of absorbing arginine from the medium and releasing proline into the environment, the amount of arginine taken from the medium was considerably greater in the GLDC27-FiPS4F-1 line (more than 2.5 times higher) than in the control line (Figure 2C). The GLDC27-FiPS4F-1 cells showed significant increases in the uptake of methionine and cysteine, as well as elevated homocysteine secretion compared to the control cells (Figure 2D). These last observations suggest reduced re-methylation due to impaired folate metabolism. Furthermore, the GLDC27-FiPS4F-1 line displayed a significant increase in the absorption of large neutral amino acids (LNAAs), consistent with higher consumption of these amino acids from the medium compared to the healthy control (Figure 2E,F). Decreased demand for aspartate, glutamate, and glutamine in the GLDC27-FiPS4F-1 line compared to the control was also observed (Figure 2G). Additionally, there was a marked increase in the absorption of glutathione from the culture medium in the GLDC27-FiPS4F-1 line (Figure 2H), suggesting a substantial decrease in its intracellular levels.

To explore the molecular underpinnings of the metabolomics data, we analyzed the expression of genes linked to serine–glycine–one-carbon metabolism (Figure 3A). Our study included a transcriptional analysis of *PHGDH* (phosphoglycerate dehydrogenase), *PSAT1* (phosphoserine aminotransferase), and *PSPH* (phosphoserine phosphatase), which are associated with the cytosolic production of serine from the glycolytic intermediate, -3-phosphoglycerate-(3-PG) (Figure 3B). We also examined the cytosolic *(SHMT1)* and mitochondrial (*SHMT2*) isoforms of serine hydroxymethyl-transferases (SHMTs) responsible for the reversible conversion of L-serine to glycine (Figure 3C). Furthermore, we assessed the cytosolic (*MTHFD1*) and mitochondrial (*MTHFD2*) isoforms of methyl-tetrahydrofolate dehydrogenase, along with the folate cycle enzymes mitochondrial *DHFR* and *MTHFR* encoding for dihydrofolate and 5,10-methylene-THF reductases, respectively. In a subsequent step, we evaluated *TYMS* (thymidylate synthetase) for thymidylate pyrimidine nucleotide synthesis (Figure 3D). The results revealed a significant reduction in the expression of *PSAT1*, *PSPH*, *SHMT2*, *MTHFD2*, *MTHFR,* and *DHFR*, while *PHGDH*, *SHMT1*, and *MTHFD1* were increased, all in comparison to the control iPSCs (Figure 3B–D). Western blot analysis at the protein level corroborated the gene expression data (Figure 3E).

### 2.3. Metabolic Alterations in GLDC27-FiPS4F-1 Impact Cell Growth and Cell Cycle Progression

In line with the contribution of serine and glycine metabolism in their synthesis, the intracellular levels of purines and pyrimidine derivatives were significantly impaired in GLDC27-FiPS4F-1 (Figure 4A). A reduction in nucleotide precursors can result in altered cell proliferation and impact cell cycle progression. Figure 4B illustrates a significant reduction in the number of the GLDC27-FiPS4F-1 cells compared to the control at all seeding densities tested. For assessing cell cycle distribution, cells in asynchronous growth were stained with propidium iodide, and flow cytometry was used to analyze their DNA content. The study revealed a noticeable difference in cell cycle distribution between the control and GLDC27-FiPS4F-1 cells. In GLDC27-FiPS4F-1, we detected a significant accumulation of cells in quiescence (sub-G1 phase), in addition to an increased proportion of cells with 4N DNA content (G2/M) and a reduction in cells with 2N DNA content (G1 phase) (Figure 4C). To further investigate the potential impact of specific cyclins on these observed differences in the cell cycle, we employed RT-qPCR to measure the expression levels of various cyclins and cyclin-dependent kinases (CDKs) that play crucial roles in cell cycle regulation (Figure 4D). The results showed a significant decrease (52%) in the expression level of *CCND1*, a gene encoding cyclin D1. Additionally, the gene expression levels of *CDK1*, *CCNA2*, and *CDK2* were all statistically significantly decreased concerning the control. These findings suggest a delayed transition from the G2/M phase back to the G1 phase in the cell cycle.

### 2.4. Mitochondria Function in GLDC27-FiPS4F-1

To explore the potential impact of nucleotide depletion on mitochondrial DNA (mtDNA) levels, we first investigated the ratio of mtDNA to nuclear (nDNA) by comparing *ND1/18S* and *12S/18S* (Figure 5A) levels measured by RT-PCR. We observed an increased ratio of mtDNA/nDNA in the GLDC27-FiPS4F-1 cells compared to the control. Consistent with this increase, we also detected higher levels of the mtDNA-translated MTCO1 protein through Western blot analysis of the OxPhos proteins (Figure 5B). Subsequently, we assessed the basal mitochondrial function of GLDC27-FiPS4F-1 and control iPSCs using Seahorse Bioscience XF96 (Figure 5C). We did not observe significant differences between the two groups regarding basal respiration or ATP production. An electron microscopy analysis of mitochondrial structures revealed a preferential presence of spherical mitochondria with poorly developed cristae in both GLDC27-FiPS4F-1 and control iPSCs (Figure 5D). In *GLDC*-deficient iPSCs, we identified a reduced population of condensed mitochondria characterized by a dense matrix and dilated cristae, a feature not observed in the control group.

### 2.5. GLDC27-FiPS4F-1 and Control iPSCs Are Efficiently Differentiated to Neural Progenitor Cells

The primary neural progenitor cells (NPCs) possess the potential to differentiate into neural and glial cell types. Thus, we initiated the neuralization process by culturing undifferentiated GLDC27-FiPS4F-1 and control iPSCs in the presence of a dual SMAD signaling inhibitor using the STEMCell Technologies monolayer protocol to generate neural progenitors. The ability of our iPSC lines to generate neural cells was evaluated by analyzing the transcription of various mRNAs. As is shown in Figure 6A–C, the gene expression analysis of iPSCs on day 0 of differentiation revealed high expression levels of the pluripotency biomarkers *OCT3/4* and *LIN28B*. These levels decreased by day 20 of differentiation, coinciding with the upregulation of neuroepithelial markers *NESTIN*, *PAX6*, and *SOX1* in both patient and control NPCs. Confocal microscopy using specific antibodies against PAX6, Nestin, and OCT3/4 corroborated the RNA results (Figure 6D). Our findings confirmed that neural induction had been successfully conducted for both cell lines. After 20 days in culture, the GCSP protein remained drastically reduced at the NPC stage in *GLDC*-deficient cells (Figure 6E). The NPCs of both groups retained their proliferative capacity but exhibited some morphological differences (Figure 6F), which deserved further evaluation. We compared the expression levels of the same markers in the GLDC27-FiPS4F-1 and the control NPC lines (Figure 6G). The GLDC27-FiPS4F-1 NPC line exhibited higher levels of *PAX6*, *NES*, and *SOX1* markers specific to the NPC stage. Additionally, it displayed *FOXG1* and *MAP2* markers specific to immature and mature neurons, respectively.

### 2.6. Astrocyte Induction from Neural Progenitors Shows Different Phenotypes

The final stage of this study involved differentiating NPC lines into human induced astrocytes (iAs) using a biphasic protocol. The differentiation process was validated by the downregulation of *PAX6* and the upregulation of a panel of cell-specific markers, including those expressed in radial glia, mature astrocytes, and neurons, which were measured on day 42 of the NPC-to-iA differentiation process (Figure 7A–C). Our data demonstrated that all markers analyzed except *PAX6* exhibited higher expression levels in iAs compared to the corresponding NPCs (Figure 7B,C). In addition, GLDC27-FiPS4F-1-iAs expressed higher levels of *GFAP*, *S100β*, *AQP4*, and *ALDH1L1* than those measured in the control iAs (Figure 7D). Moreover, confocal analysis following immunostaining with GFAP and S100β showed a significant difference in the morphology and frequency of positive cells between the groups. The control culture displayed strong S100β labeling, while GFAP staining was weak. The analysis of patient-derived iAs revealed a population of cells that were morphologically heterogeneous and smaller in size than the control group (Figure 7E). The cells displayed a star-shaped morphology with long and numerous projections and exhibited positive staining for S100β and GFAP, although not always colocalized (Figure 7F). Cells expressing GFAP resembled radial glia (Figure 7F). The quantification of *MAP2* and *NeuN* expression levels confirmed a significant increase in the expression of mature neuronal markers in GLDC27-FiPS4F-1iAs compared to control iAs (Figure 8A). Confocal analysis corroborated the RT-qPCR result (Figure 8B).

The functional maturity of the astrocytes produced was subsequently examined. A key indicator of astrocyte function is its Na^+^-dependent glutamate uptake, which reflects the activity of GLAST and GLT-1 transporters. This function can be directly measured using a radioactive assay. The presence of both transporters was evaluated using RT-qPCR and immunostaining of GLAST and GLT-1 (Figure 8C,D). The data revealed that 6-week-old iPSC-derived astrocytes were capable of glutamate uptake, albeit with varying efficiencies. GLDC27-FiPS4F-1 iAs, with the highest levels of GLAST protein expression, exhibited the most robust transport activity (Figure 8E). This was corroborated by a significant reduction in extracellular glutamate levels in GLDC27-FiPS4F-1 iAs compared to the control cells (Figure 8F). 

In summary, the differentiation of the two NPC lines yielded distinct iA cultures, differing in the presence of key markers and functional capacity, as evidenced by their glutamate transport capabilities.

### 2.7. Maintenance of Glycine Homeostasis in GLDC27-FiPS4F-1 iAs

After confirming the absence of GCSP protein in GLDC27-FiPS4F-1 iAs (Figure 9A), we proceed to compare the amino acid levels with those detected in the control iA culture and the corresponding iPSCs. Notably, there was a significantly higher uptake of glycine from the medium in GLDC27-FiPS4F-1 iAs compared to the control iAs (Figure 9B). Consistent with this finding, the expression levels of the glycine transporters *GLYT1* and *GLYT2* were, respectively, 1.5 and 4 times higher in the GLDC27-FiPS4F-1 iAs than in the control (Figure 9C). Subsequently, we examined the expression levels of representative genes (Figure 9D) to assess the flux through alternative metabolic pathways that could help maintain physiological glycine levels. No significant alterations in serine uptake from the medium profile were detected in *GLDC*-deficient iAs (Figure 9E). We measured the expression of the *SRR* gene, which encodes serine racemase, to estimate the synthesis of D-serine from L-serine. The expression of this gene consistently increased during the differentiation process. However, at the astrocyte stage, the expression level in patient-derived cells was ten times higher than in the control iAs (Figure 9F). On the contrary, the expression of the *SHMT2* gene was impaired in the *GLDC*-deficient iPSCs, NPCs, and iAs (Figure 9G). These results were further confirmed at the protein level **(**Figure 9H). Additionally, we observed increased demand for arginine from the medium observed in GLDC27-FiPS4F and GLDC27-FiPS4F-1 iAs (Figure 9I). This finding was concordant with a simultaneous increase in *AGAT* gene expression (Figure 9J,K), which encodes for L-arginine:glycine amidino transferase, in the *GLDC*-deficient iPSCs, NPCs, and iAs.

## 3. Discussion

The primary aim of this study was to establish a valid human model of *GLDC* deficiency to serve as a basis for testing potential therapeutic strategies in nonketotic hyperglycinemia (NKH). We generated a human iPSC line that exhibited biochemical characteristics of *GLDC* deficiency while maintaining pluripotency, the capability to differentiate into three germ layers, and genomic stability [15]. The GLDC27-FiPS4F-1 line showed a noteworthy decrease in the amounts of GCSP protein and the assembled GCS complex. The decarboxylase activity in this line was only 15% of that in the control line. Due to the impaired ability of the NKH line to catabolize glycine, there was an accumulation of this amino acid in the culture medium. In contrast, the control line required the uptake of glycine. We hypothesize that this remaining GCS activity enables the cells to survive and develop adaptive mechanisms to compensate for the reduced enzyme activity. 

The impact of *GLDC* deficiency on cellular metabolism was evaluated by quantifying specific sets of metabolites and proteins. Previous studies have reported disturbances in the serine shuttle and metabolism in NKH patients [18]. In GLDC27-FiPS4F-1, we observed increased demand for serine, methionine, and cysteine from the extracellular milieu, which could be understood in the context of the cellular response to maintain 5,10-methylen-THF. We investigated whether this specific metabolite pattern could be explained by the down- or upregulation of specific metabolic pathways related to the serine–glycine–one-carbon metabolism. The “de novo” process of serine synthesis begins with 3-phosphoglycerate (3-PG), which is then sequentially processed by the enzymes PHGDH, PSAT1, and PSPH. Our findings indicate a significant increase in *PHGDH* expression, accompanied by a decrease in *PSAT1* and *PSPH* gene expression in the patient-derived line. The main function of PHGDH in proliferative cells is to maintain folate stores intended for nucleotide synthesis [19]. Consistent with the significant reduction in NADH production via GCS flux, GLDC27-FiPS4F-1 cells seemed to process 3-PG to obtain NADH but reduced the expression of the other enzymes in the pathway, resulting in poor L-serine synthesis. This could justify the increase in serine uptake from the medium. We also detected impairment in mitochondrial *SHMT2* and *MTHFD2* and increases in cytosolic *SHMT1* and *MTHFD1*. These variations from normal flux may reflect a compensatory cytosolic mechanism to reverse the loss of mitochondrial one-carbon metabolism, like that described in [20] for proliferating mammalian cell lines. Our study found that despite the metabolic shift to meet the proliferative demand of the iPSCs, there was a concomitant alteration in cell growth and cell cycle progression, with a significant number of cells in quiescence. This was consistent with the decreased demand for aspartate and a nearly 50% decline in the levels of nucleotide metabolites, as described in [21]. Furthermore, in line with earlier findings in mouse models and human cells [22], the culture medium from GLDC27-FiPS4F-1 exhibited a substantial decrease in glutathione levels, probably implying a greater necessity to counteract ROS production. 

The data support metabolic rewiring at the pluripotent stage in GLDC27-FiPS4F-1, which is necessary but probably insufficient to meet the changing biosynthetic demands at different cell cycle stages and could influence cell fate during the differentiation process. The expression of GCS in radial glia has been proposed as a relevant factor in the proper development of the cerebral cortex [23]. We evaluated the capacity of iPSCs to differentiate into astrocytes through an intermediate stage of neural progenitors capable of producing neurons and glial cells. 

The loss of *OCT3/4* and *LIN28B* markers, accompanied by the gain of *PAX6*, *NES*, and *SOX1* expression compared to the iPSC stage, confirmed that both cell lines correctly differentiated to the NPC stage [24]. Differences in the morphology and expression of neuroepithelial and neuronal lineage markers, which appeared upregulated, were observed in the patient-derived NPC line. This observation may indicate premature senescence and differentiation, in agreement with the reduced cell viability and proliferative capacity measured in patient-derived iPSCs [11,25]. To characterize iAs, we evaluated an extensive set of astrocytic markers, including *GFAP*, *S100ß*, *ALDH1L1*, *AQP4*, *APOE*, *GLAST*, and *GLT-1* [26,27,28]. All markers exhibited significantly higher expression in iAs compared to NPCs, confirming their higher degree of astroglial differentiation. However, we noted differences in the morphology and relative expression of specific markers between control and patient-derived iAs. The control group displayed a quiescent morphology, characterized by the reduced expression of *GFAP* and *GLAST*, and increased levels of *GLT-1*, all of which are typical of mature astrocytes [29]. In contrast, the iAs derived from the patient showed a fibrous or radial morphology. Relative to the control group, these cells exhibited reduced *GLT-1* expression and significant increases in *GFAP* and *GLAST* expression. Such changes have also been observed in radial glia and astrocytes [30], supporting the notion that a considerable proportion of radial glial cells are present in GLDC27-FiPS4F-1 iAs [31,32]. Elevated *GFAP* expression has also been linked to reactive phenotypes associated with pathological conditions [26,32,33]. Furthermore, we observed increased expression of the neuronal markers *MAP2* and *NeuN* in GLDC27-FiPS4F-1 iAs, indicating the presence of mature neurons in this culture. A ten-fold increase in *SRR* gene expression detected in the GLDC27-FiPS4F-1 iA culture corroborated these findings [34]. Additionally, the GLDC27-FiPS4F-1 iA culture exhibited significant increases in glutamate and glycine uptake, along with elevated expression of the glycine transporters (*GLYT1* and *GLYT2*) [35]. Considering the presence of neurons, these changes may represent a regulatory response to maintain neurotransmitter balance in an active neural network, as both amino acids serve as neurotransmitters in the central nervous system [36]. 

Finally, we hypothesized that *GLDC*-deficient cells would activate metabolic pathways to re-establish glycine homeostasis. We observed a significant increase in arginine absorption, accompanied by substantial upregulation in *AGAT* gene expression at both the iPSC and iA stages. This pattern is compatible with the increased synthesis of guanidinoacetate, as described in the brains of adult *GLDC*-deficient mice [37]. Such upregulation could adjust the intracellular glycine concentration by enhancing the synthesis of guanidinoacetate, acting as an ‘exit strategy’. 

## 4. Materials and Methods

### 4.1. Cell Culture Conditions

Healthy CC2509 fibroblasts (Lonza, Basel, CHE) and patient-derived fibroblasts were cultivated according to standard procedures and used before reaching 10 passages. In brief, the cells were maintained in Minimum Essential Medium (MEM) (Sigma-Aldrich, St. Louis, MO, USA) supplemented with 1% (*v*/*v*) glutamine, 10% fetal bovine serum (FBS), and a 0.1% antibiotic mixture (penicillin/streptomycin) under standard cell culture conditions (37 °C, 95% relative humidity, 5% CO_2_).

### 4.2. iPSC Characteristics

One healthy control iPSC line (registered as FiPS Ctrl2-SV4F-1), matched for age, sex, ethnicity, and reprograming method with the GLDC27-FiPS4F-1 line used in this study, was obtained from the Banco Nacional de Líneas Celulares of the Instituto de Salud Carlos III. The GLDC27-FiPS4F-1 line was reprogrammed at our laboratory [15] from NKH patient-derived fibroblasts who suffered a severe clinical course and bore the biallelic changes c.1742C > G (p.Pro581Arg) and c.2368C > T (p.Arg790Trp) in the *GLDC* gene that led to a significant decrease in protein levels and exchange activity in vitro, as previously described [16]. These were then registered in the Banco Nacional de Líneas Celulares of the Instituto de Salud Carlos III as GLDC27-FiPS4F-1.

### 4.3. Genetic Background of the iPSCs Lines 

Prior to beginning iPSC differentiation, we identified potential pathogenic nucleotide sequence variations in both cell lines using array CGH and whole-exome sequencing. Variants from over 2000 genes were filtered based on minor allele frequency (MAF), while also considering commonly occurring mitochondrial DNA mutations in iPSCs clones. Annex 1 (Appendix A) presents a summary of the identified variations in both cell lines.

### 4.4. Sample Collection and Metabolite Analysis 

Harvested after two days of culture in mTeSR, six-well plates of each cell line, as described [38], were counted. Aliquots of the spent-culture medium and the corresponding cells, without any residual medium, were collected separately and immediately frozen in liquid nitrogen and stored at −80 °C until analysis. Blank culture medium samples were also collected and stored under the same conditions. Both medium and cell extracts were processed and deproteinized before measurement. Amino acids were measured by Ionic Exchange Chromatography (IEC). Purines and pyrimidines were determined by High-Performance Liquid Chromatography (HPLC) according to [39]. Total homocysteine, quantified after derivatization with SBD-F by HPLC with fluorescence detection, was measured as described in [40,41]. The results of the spent-culture medium represent the consumption or excretion values of the metabolite using the baseline value measured in the blank medium. Raw data were in pmol/µg protein/72 h. Total protein concentrations were measured using Bradford’s assay.

### 4.5. Oxygen Consumption Rate (OCR) Evaluation

Cellular Consumption Respiration (OCR) was analyzed using the Flux Analyzer XFe96 (Agilent-Seahorse) device. We seeded iPSCs or NPCs on Matrigel-coated Seahorse XF96 cell plates. The test was carried out using the Seahorse XF Cell Mito Stress Test Kit (Agilent). Before measurements, cells were washed and incubated for 1 h at 37 °C in CO_2_-free conditions in XF DMEM Medium, pH 7.4 (Agilent), previously warmed and supplemented with pyruvate, glucose, and glutamine at final concentrations of 1mM, 10mM, and 2 mM, respectively. Subsequently, various drugs were sequentially injected to reach the following final concentrations: oligomycin (1.5 μM), carbonyl-cyanide-p-trifluoro-methoxy phenylhydrazone (FCCP 20 μM), and a combination of rotenone/antimycin A (0.5 μM each). FCCP concentration was previously optimized through titration. Following the completion of the measurement, all wells in the p96 plate were stained with Hoechst (1:100) for 30 min at room temperature. The number of cells per well was quantified for data normalization using the Cytation 5 image reader. The data obtained were processed using software provided by the manufacturer: seahorseanalytics.agilent.com (accessed on 20 October 2022). 

### 4.6. Cell Viability and Cycle Progression

Cell viability assays were conducted 72 h after plating iPSCs seeded at varying densities (10,000; 20,000; and 40,000 cells per well) using the CellTiter 96 Aqueous One Solution Cell Proliferation kit^®^ on 96-well plates, following to the manufacturer’s protocol. For cell cycle analysis via flow cytometry, iPSCs prepared as described in [34] were treated with Propidium Iodide/RNase Staining Buffer (BD Pharmingen, San Diego, CA, USA) for 30 min at 37 °C in darkness. The data were collected using a BD FACSCanto II flow cytometer (BD Biosciences, San Jose, CA, USA) and FACSDiva 8.0 software. The relative percentages of cells in the sub-G1/G1, S, and G2/M phases of the cell cycle were determined using FlowJo v.2.0 software. 

### 4.7. RT-qPCR 

Total RNA was extracted using TRIzol and subsequently converted to cDNA with an NZY First-Strand cDNA Synthesis (NZYTech) kit (NZYTech, Lda. Lisboa, Portugal). Amplification was performed using Perfecta SYBR Master Mix (Quantabio) on a LightCycler 480 (Roche Applied Science, Indianapolis, IN). Primers were designed by using the Primer3 v.0.4.0 software. For data normalization, three different genes—*ACTB*, *GAPDH*, and *GUSB*—were initially analyzed, with *ACTB* proving to be the most stable. Relative quantification was carried out using the standard 2^−ΔΔCt^ method. See Appendix A for primers sequences.

### 4.8. Mitochondrial DNA Content 

Mitochondrial DNA (mtDNA) content was calculated using quantitative RT-PCR by measuring the threshold cycle ratio (ΔCt) of the mitochondria-encoded genes mtDNA *ND1* and *12S* and the nuclear *18S*. Data were expressed as mtDNA/nuclear DNA (nDNA).

### 4.9. Enzymatic Activity 

GCSP protein enzymatic activity was determined in triplicate using the exchange reaction between radio-labeled bicarbonate NaH_14_CO_3_ and glycine as described in [16].

The results were normalized to protein levels measured using the Lowry method.

### 4.10. Measurement of Protein Levels using SDS Western Blot 

Cell lysates were prepared using a lysis buffer (2% Triton X-100, 10% glycerol, 150 mM NaCl, 10 mM Tris–HCl pH 7.5, 150 mM NaCl) and subjected to freeze–thaw cycles. The resulting supernatants were used for Western blotting, with protein concentration determined using the Bradford method (Bio-Rad Laboratories, Hercules, CA, USA). Electrophoretic separation was performed using two different systems: NuPAGE gradient gels of 4–12% acrylamide or manually prepared 10% acrylamide/bisacrylamide gels. Both systems employed Lonza’s ProSieve™ Color Protein Markers as molecular weight standards. Following electrophoresis, gels were transferred onto 0.2 mm nitrocellulose membranes using the iBlot Gel Transfer Stacks Nitrocellulose Regular system. In all studies, detection was achieved using a secondary horseradish peroxidase-conjugated antibody, followed by membrane development with ECL (GE Healthcare, Piscataway, NJ, USA). Band intensities were quantified using a BioRad G-8000 scanner. The specific antibodies used in this study are listed in Appendix A.

### 4.11. Protein Complex Analysis in Native Conditions 

The analysis of the GCS complex was conducted using the NativePAGE™ Novex^®^ Bis-Tris Gel System (Invitrogen, Carlsbad, CA, USA). Cell sediments were resuspended in a mixture of 2% digitonin, 4× Sample Buffer, and dH_2_O and incubated for 15 min at 4 °C. This was followed by centrifugation at 20,000× *g* for 15 min to collect the supernatant, whose protein concentration was measured using the Bradford method. Prior to loading onto NativePAGE™ Novex 3–12% gels, G-250 additive was added to the samples. The electrophoretic separation was carried out in two phases. After separation, the gels were washed and incubated with 2× transfer buffer, and then, transferred to PVDF membranes with the iBlot™ 2 Gel Transfer Device (Invitrogen, Waltham, MA, USA). This was followed by incubation with 8% acetic acid and subsequent washing with 100% methanol. Finally, the immunodetection procedure was performed using primary and secondary antibodies, as detailed in Appendix A.

### 4.12. Immunofluorescence Staining 

For immunofluorescence staining, cells were fixed in 10% formalin for 20 min at room temperature, washed with 0.1% PBS-Tween, and permeabilized with PBS/0.1% Triton X-100 for 10 min. The cells were then incubated in a blocking solution for a minimum of 30 min. Following this, cells were incubated overnight at 4 °C with primary antibodies at appropriate concentrations (Appendix A). Next, secondary antibodies, already conjugated to fluorophores (ThermoFisher, Waltham, MA, USA), were incubated in the blocking solution at the appropriate concentrations for 30 min. To stain cell nuclei, DAPI at a dilution ratio of 1:5000 (Merck, Rahway, NJ, USA) was used. The cells were then mounted using Prolong Diamond Antifade (ThermoFisher). The samples were examined using an Axiovert200 inverted microscope (Zeiss, Jena, Germany) equipped with GFP, DsRed, and Cy5 fluorescence filters (10×, 25×, and 40×). Image visualization and fluorescence quantification were carried out using the ImageJ-FiJi (Java 8) software. 

### 4.13. Differentiation to Neural Progenitor Cells (NPCs) and Induced Astrocytes (iAs) 

Both Ctrl2-SV4F-1 and GLDC27-FiPS4F-1 iPSC lines underwent differentiation into Neural Progenitor Cells (NPCs) following the supplier’s instructions. We implemented rigorous standardization of iPSC reprogramming and differentiation. Both cell lines were matched for time in culture and passages. Cells plating densities were optimized to prevent potential alterations due to local growth factors. A panel of cell markers was used to assess cell types along the differentiation pathway, and the quality of cultures was monitored by evaluating cellular functionality.

Briefly, iPSCs were seeded at a density of 250,000 cells per well in Matrigel-coated 6-well plates (Corning, Corning, NY, USA) using mTesR plus (STEMCell Technologies, Vancouver, BC, Canada) containing 10 μM ROCK inhibitor Y-27632 (STEMCell Technologies). On day 3, the medium was switched to a STEMdiff^TM^ SMADi Neuronal Induction Kit (STEMCell Technologies) and refreshed daily. By day 8, the cells were replated at a density of 250,000 cells/cm^2^ on Matrigel-coated 6-well plates. This process was repeated twice more. After the third passage, the culture medium was changed to STEMdiff^TM^ Neural Progenitor Medium, with daily replacements. At this differentiation stage, cells were analyzed using RT-qPCR and immunofluorescence for expression of the NPC markers *SOX1*, *PAX6*, and *NES*. Cell banks in this passage were cryopreserved in STEMdiff^TM^ Neural Progenitor Freezing Medium, employing a slow freezing protocol (approximately 1 °C/min reduction) according to the supplier’s guidelines. 

For astrocyte differentiation, cryopreserved NPCs were thawed in Matrigel-coated 6-well plates. Once 80–90% confluence was reached, NPCs were harvested using Acutase^TM^ (Millipore, Burlington, MA, USA) and plated at a density of 2 × 10^5^ cells/cm^2^ in STEMdiff^TM^ Astrocyte Differentiation Medium (STEMCell Technologies) in Matrigel^®^ (Corning)-precoated 6-well plates. After seven days in culture, with daily medium changes, the first cell colony passage was performed. The cells were dissociated using Acutase^TM^ (Millipore) and seeded at 1.5 × 10^5^ cells/cm^2^ in new Matrigel^®^-treated 6-well plates (Corning). This process was repeated twice. From the second passage (day 14 of the differentiation process), the culture medium was changed every 48 h. Following the third passage, the culture medium was switched to STEMdiff^TM^ Astrocyte Maturation. After three further passages in STEMdiff^TM^ Astrocyte Maturation, the developed cell line was characterized through confocal microscopy, RT-qPCR analysis of specific markers, and functional evaluation of glutamate transport. Additionally, expression analysis of proteins related to NKH pathology, serine–glycine–one-carbon metabolism, and metabolite measurement in the culture medium was conducted at this stage. 

### 4.14. Glutamate Uptake 

Glutamate transport was quantified using previously established methods [42]. In brief, 2.5 × 10^5^ iAs cells underwent preincubation with 0.5 mL of HEPES-buffered saline for 10 min, followed by incubation in 250 µL HEPES-buffered saline containing [U-^14^C]-glutamate (0.1 pCi) for 15 min. Subsequently, cells were rinsed with 2 × 0.5 mL fresh HEPES-buffered saline (2–4 °C) within 5 s and dissolved in 250 µL of 0.2 M NaOH. A 150 µL sample was placed in micro vials and measured for radioactivity using a liquid scintillation counter. The collected data were normalized to the protein content measured through Bradford’s assay.

### 4.15. Statistical Analysis 

The statistical significance was obtained using a two-tailed Student’s t-test performed with the GraphPad Prism 6 program. Differences were considered significant at *p* values of * <0.05; ** <0.01; and *** <0.001. The GraphPad Prism 6 program and BioRender were used for images.

## 5. Conclusions

Our results indicate that *GLDC*-deficient iPSCs exhibit high resilience and survival capacity through metabolic adjustments or ‘rewiring’. The analysis of *GLDC*-deficient iPSCs demonstrates that the cells exhibit biochemical hallmarks of the disease. Additionally, they display apparent metabolic rewiring that targets amino acids and metabolites other than glycine to compensate for a potential one-carbon subunit deficiency, and thus, can be used as a pluripotent cellular model to test potential therapeutic strategies for nonketotic hyperglycinemia. However, *GLDC* deficiency results in lower cell viability and proliferative capacity, leading to cellular quiescence of the iPSCs and the irregular differentiation of induced pluripotent stem cell lines into induced astrocytes (iAs). This, in turn, leads to a heterogeneous culture in which pathways involved in the clearance of extracellular levels of glutamate and glycine are activated. This observation supports previous findings in pluripotent stem cells from mice, which indicated that the activation of GCS through *GLDC* is a crucial regulator of cell fate [11,43,44,45] hampering its use as a neural model of the disease. Although this study may offer new insights into the neurological features of nonketotic hyperglycinemia, further research is necessary.

## Figures and Tables

**Figure 1 ijms-25-02814-f001:**
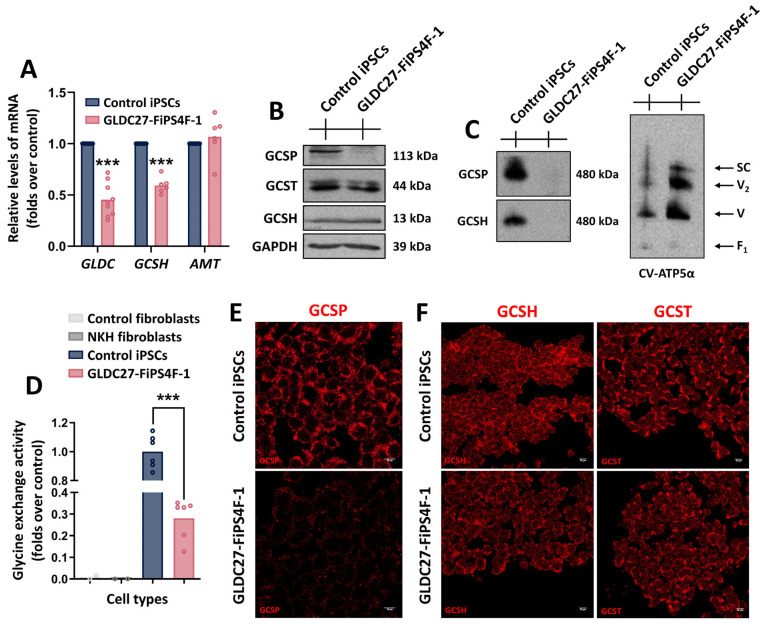
GCS complex in the GLDC27-FiPS4F-1 line. (**A**) Relative quantification of *GLDC*, *GCSH*, and *AMT* gene expression in GLDC27-FiPS4F-1 line related to control iPSCs. Standardization was performed using the endogenous *ACTB* gene. Data represent the average of *n* = 3 biological replicates conducted in triplicate. (**B**) Representative SDS-PAGE Western blot of GCSP, GCSH, and GCST proteins in iPSCs lines. GAPDH was used as the loading control. (**C**) Supramolecular structure analysis using native gels under non-denaturing conditions and GCSP or GCSH antibodies. CV-ATP5α was used as the loading control. (**D**) Glycine exchange enzyme activity evaluation in control and NKH patient fibroblasts as well as in control and GLDC27-FiPS4F-1 iPSC lines. Data represent the average of *n* = 2 biological replicates conducted in triplicate. (**E**,**F**) Immunofluorescence staining and laser scanning confocal imaging of GCS complex proteins in control and GLDC27-FiPS4F-1 iPSCs. Scale bar: 10 μm. 40× magnification. Data represent the average of *n* = 3 biological replicates conducted in triplicate. (**A**,**D**) Statistical analysis: Student’s *t*-test (*** *p* < 0.001).

**Figure 2 ijms-25-02814-f002:**
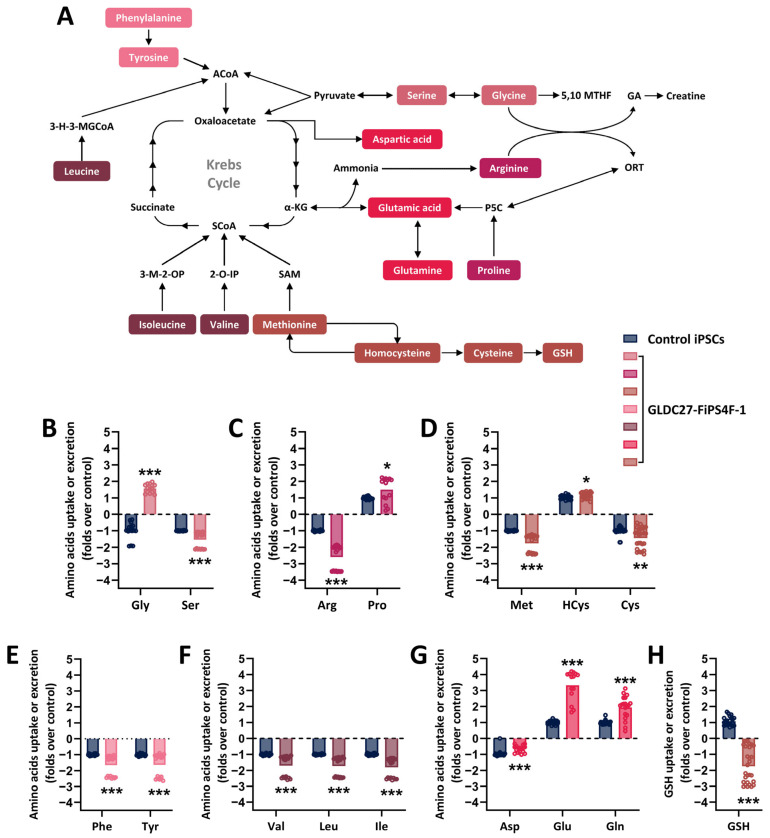
Relative quantification of amino acids and GSH in the extracellular medium after 72 h of culture. (**A**) Simplified overview of amino acid metabolism in the cell. The colors represent different amino acid metabolic groups as defined by the KEGG platform: https://www.genome.jp/kegg/pathway.html#amino (accessed on 11 January 2023). (**B**–**H**) Metabolite measurements were performed on *n* = 20–30 samples in iPSCs lines that had undergone 3–6 passages post-thawing. The graphs illustrate the GLDC27-FiPS4F-1 lines’ (different pink, red, and brown shades) capacity for the uptake or secretion of various metabolites compared to that of the control iPSCs line (blue). The control line is assigned a value of 1 or −1, indicating whether it expels (1) or absorbs (−1) the metabolite from the medium. The dotted line represents the 0 value. All values obtained in the measurement were normalized by protein. Baseline values of each metabolite in the medium were used to determine the iPSCs lines’ behavior (either uptake or secretion). Statistical analysis: Student’s *t*-test (* *p* < 0.05; ** *p* < 0.01; *** *p* < 0.001). ACoA: acetyl-CoA; 3-H-3-MGCoA: 3-hydroxy-3-methylglutaryl-CoA; SCoA: succinyl-CoA; α-KG: α -ketoglutarate; 3-M-2-OP: 3-methyl-2-oxopentanoate; 2-O-IP: 2-amino-3-ketobutyrate; SAM: S-adenosylmethionine; 5,10 MTHF: 5,10-methylenetetrahydrofolate; P5C: pyr-roline-5-carboxylate; GA: guanidinoacetic acid; ORT: ornithine; GSH: glutathione.

**Figure 3 ijms-25-02814-f003:**
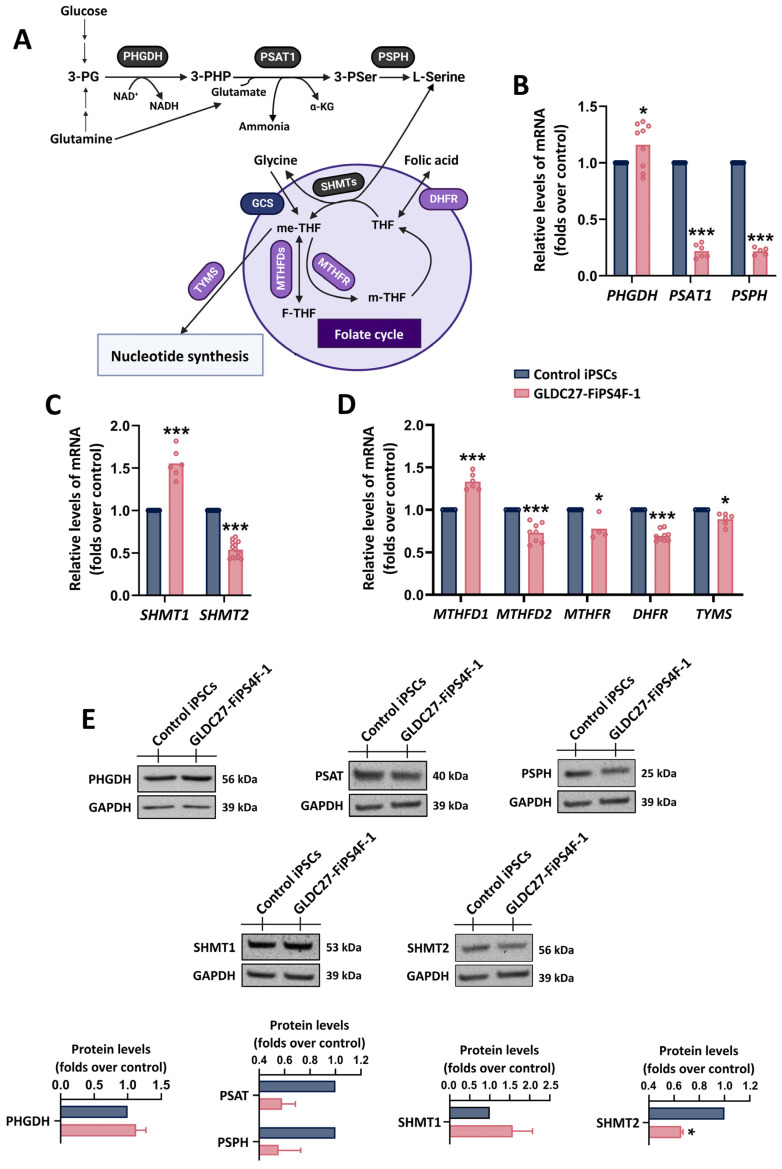
Serine–glycine–one-carbon metabolism. (**A**) Diagrammatic representation of key enzymes involved in serine-glycine metabolism and folate cycle. Relative quantification by RT-qPCR of genes (**B**) related to the synthesis of serine from glycolytic intermediates, (**C**) related to the reversible synthesis of glycine from serine, and (**D**) involved in one-carbon metabolism (folate cycle). Data were standardized against the endogenous *ACTB* gene. Data represent the average of *n* = 3 biological replicates conducted in triplicate. (**E**) Representative Western blot and protein quantification using GAPDH as loading control. Data represent the average of *n* = 2 biological replicates. (**B**–**E**) Statistical analysis: Student’s *t*-test (* *p* < 0.05; *** *p* < 0.001).

**Figure 4 ijms-25-02814-f004:**
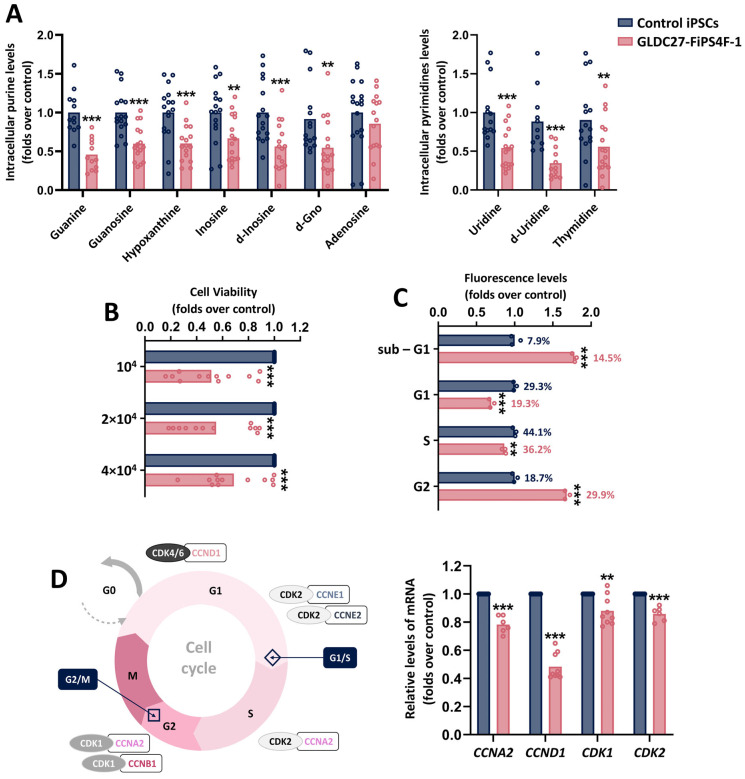
Nucleotide synthesis, viability, and cell cycle of iPSCs lines. (**A**) Relative quantification of intracellular purines and pyrimidine derivative levels in control and GLDC27-FiPS4F-1 extracts after 72 h of culture. Nucleotide measurements were performed on *n* = 17 iPSCs samples that had undergone 3–6 passes post-thawing. (**B**) Relative quantification of GLDC27-FiPS4F-1 cell density at three different platting densities compared to the control line measured after 72 h of culture. (**C**) Analysis of cell cycle distribution by flow cytometry. On bars, percentages represent the fraction of cells in each cycle stage. Data represent the average of *n* = 3 biological replicates conducted in triplicate. (**D**) Schematic representation and relative quantification of expression of cyclin-coding genes and CDK-coding genes involved in cell cycle regulation. Data represent the average of *n* = 2 biological replicates conducted in triplicate and were standardized against the endogenous *ACTB* gene. (**A**–**D**) Statistical analysis: Student’s *t*-test (** *p* < 0.01; *** *p* < 0.001).

**Figure 5 ijms-25-02814-f005:**
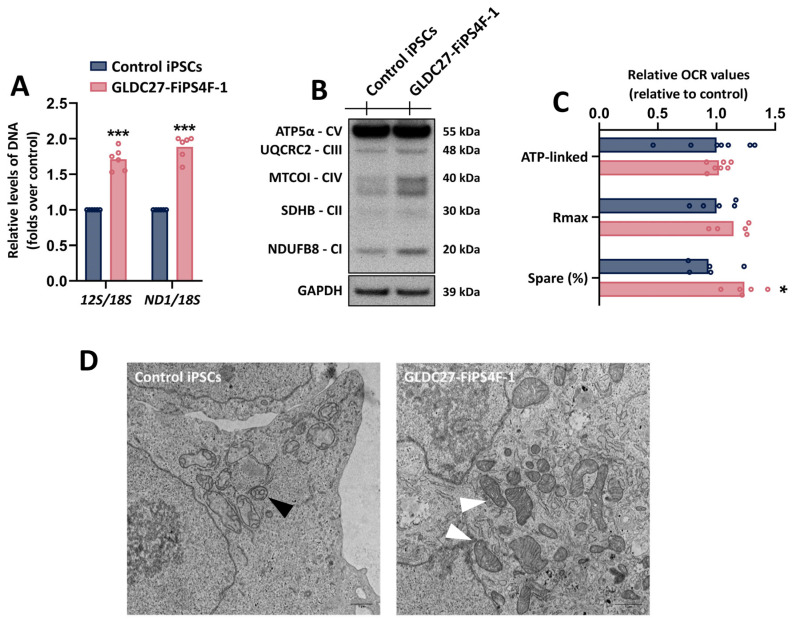
Mitochondrial function of GLDC27-FiPS4F-1. (**A**) Mitochondrial DNA depletion analysis through evaluation of *12S/18S* and *ND1/18S* ratios measured by RT-PCR. Data represent the average of *n* = 2 biological replicates conducted in triplicate. (**B**) Representative SDS-PAGE Western blot of OxPhos proteins using an antibody cocktail against different proteins of these complexes. (**C**) Mitochondrial respiration in GLDC27-FiPS4F-1 compared to control iPSCs. Graphs represent the respiratory parameters derived from oxygen consumption (OCR). Rmax: maximal respiration; Spare: spare capacity; ATP-linked: ATP production. Data represent the average of *n* = 3 biological replicates with measurements acquired 20 times. (**D**) Mitochondrial morphology analyzed by transmission electron microscopy in control (left panel) and GLDC27-FiPS4F-1 (right panel) iPSCs. Mitochondria are indicated by black and white arrowheads. (**A**,**C**) Statistical analysis: Student’s *t*-test (* *p* < 0.05; *** *p* < 0.001).

**Figure 6 ijms-25-02814-f006:**
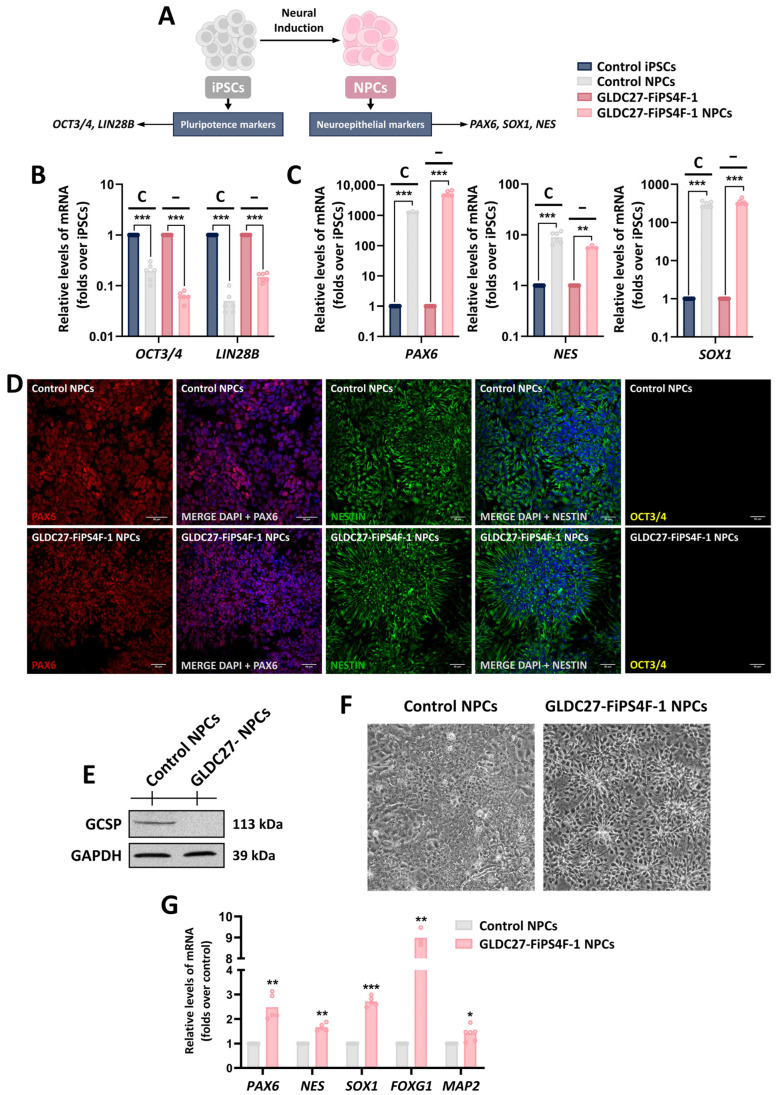
Characterization of the generated NPC lines. (**A**) Schematic representation of characteristic markers in iPSCs’ differentiation to NPCs. (**B**,**C**) Relative gene expression of pluripotency markers (*OCT3/4*, *LIN28B*), early neuroepithelium markers (*SOX1*), and neuroepithelium and radial glial markers (*PAX6*, *NES*). Graphs show gene expression levels in both control (C) and *GLDC*-deficient (−) lines compared to iPSC lines on days 0 (iPSCs) and 20 (NPCs) of differentiation. (**D**) Immunofluorescence staining and laser scanning confocal imaging of PAX6, Nestin, and OCT3/4 markers in control and GLDC27-FiPS4F-1 NPCs. Dapi: nuclear marker (blue). Scale bar: 50 μm. Magnifications of 10×. (**E**) Representative Western blot showing GCSP protein levels in NPC cultures. “GLDC27-NPCs” refers to the GLDC27-FiPS4F-1 NPC line. (**F**) Representative images of NPC cultures using phase contrast microscopy. (**G**) Relative gene expression levels of neural lineage markers in GLDC27-FiPS4F-1 NPCs compared to control NPCs. (**B**,**C**,**G**) Data represent the average of *n* = 2 biological replicates conducted in triplicate standardized against the endogenous *ACTB* gene. Statistical analysis: Student’s *t*-test (* *p* < 0.05; ** *p* < 0.01; *** *p* < 0.001).

**Figure 7 ijms-25-02814-f007:**
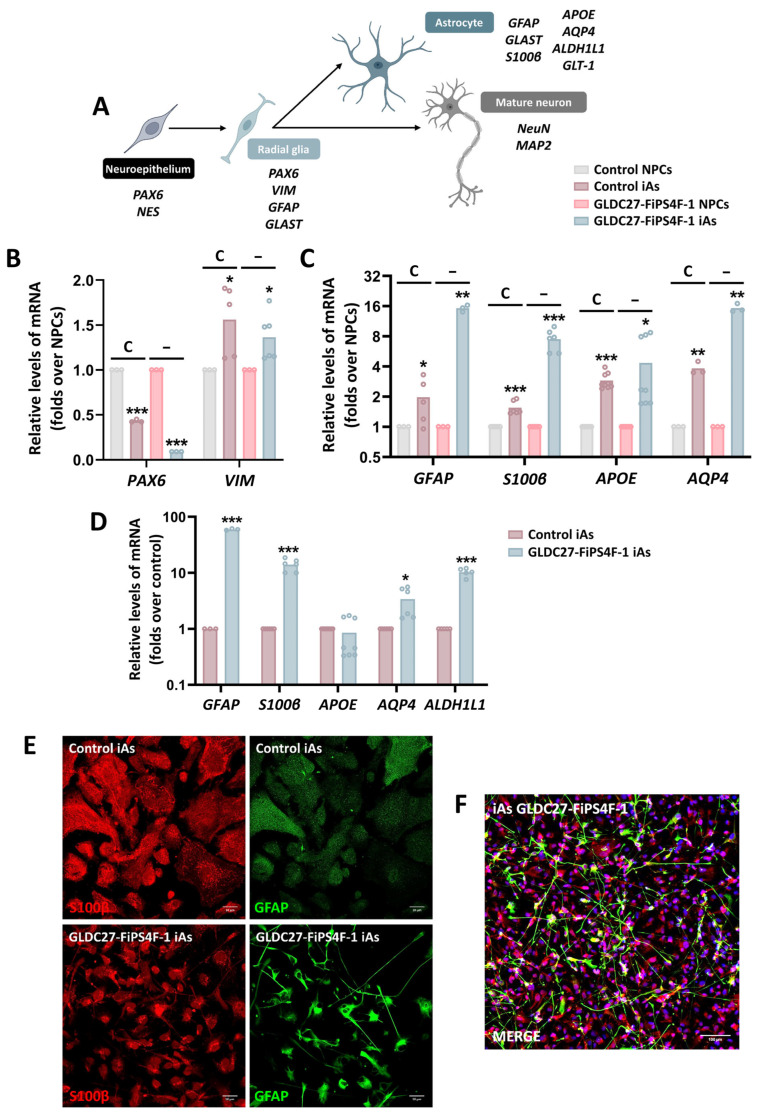
Characterization of the generated iA cultures. (**A**) Schematic diagram showing the differentiation stages of various cell types derived from neuroepithelial cells and the respective markers expressed in each stage. (**B**) Relative quantification of gene expression of radial glial markers (*PAX6* and *VIM*) and (**C**) astrocyte markers (*GFAP*, *S100β*, *APOE*, and *AQP4*). (**B**,**C**) Graphs show gene expression levels of these markers in both control (C) and *GLDC*-deficient (−) iA cultures compared to NPC lines. (**D**) Relative quantification of astrocyte marker (*GFAP*, *S100β*, *APOE*, *AQP4*, and *ALDH1L1*) expression levels in GLDC27-FiPS4F-1 iAs related to control iAs. (**E**,**F**) Immunofluorescence staining and laser scanning confocal imaging of GFAP and S100β markers. Dapi: nuclear staining (blue). (**E**) Scale bar: 10 μm. Magnifications of 40×. (**F**) Scale bar: 100 μm. Magnifications of 25×. (**B**–**D**) Data represent the average of *n* = 2 biological replicates conducted in triplicate and were standardized against the endogenous *ACTB* gene. Statistical analysis: Student’s *t*-test (* *p* < 0.05; ** *p* < 0.01; *** *p* < 0.001).

**Figure 8 ijms-25-02814-f008:**
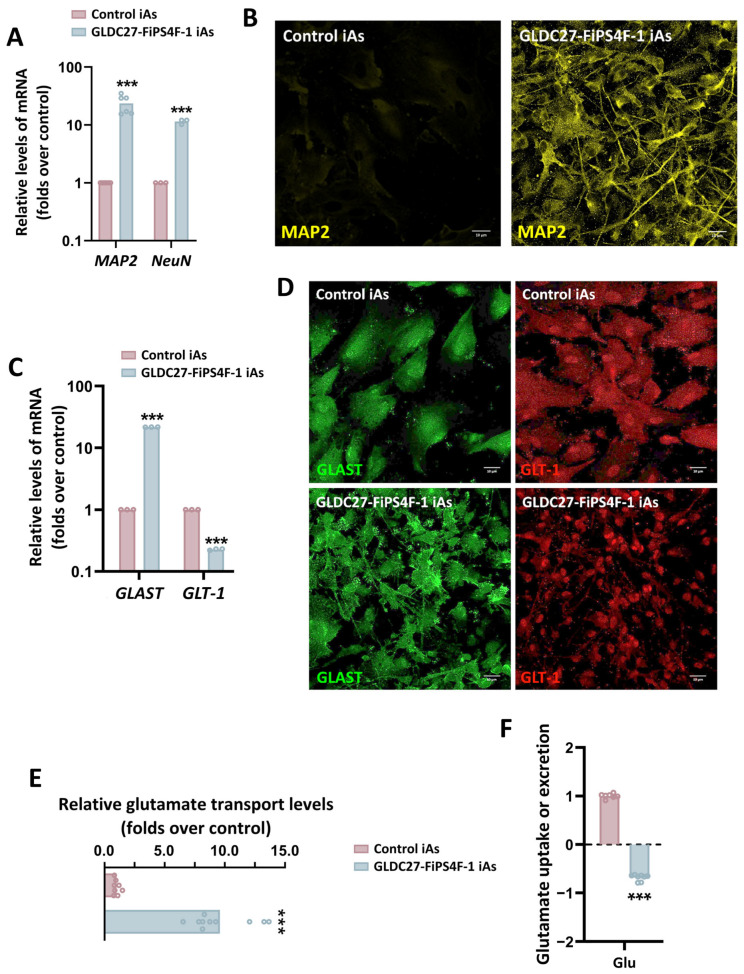
Presence of mature neurons and functional analysis of the iA cultures. (**A**) Relative quantification of gene expression levels of mature neuron markers *MAP2* and *NeuN* in *GLDC*-deficient iAs compared to control iAs. (**B**) Immunofluorescence staining and laser scanning confocal imaging of neuronal marker MAP2 in both iA cultures. (**C**) Relative gene expression levels of *GLAST* and *GLT-1* in GLDC27-FiPS4F-1 iAs related to control iAs. (**D**) Immunofluorescence staining and laser scanning confocal imaging of GLAST and GLT-1 markers in control and GLDC27-FiPS4F-1 iA cultures. (**E**) Radiolabeled glutamate transport measurement in GLDC27-FiPS4F-1 iAs compared to control iAs. Data were evaluated in two complete differentiation processes in quintuplicate. (**F**) Relative quantification of glutamate extracellular levels in control and GLDC27-FiPS4F-1 iAs after 72 h of culture. Control iAs were assigned a value of 1. The dotted line represents the 0 value. All values obtained in the measurement were normalized by protein content. A total of *n* = 10 samples were evaluated. (**A**,**C**) Data were standardized against the endogenous *ACTB* gene. Data were analyzed in triplicate (**B**,**D**) Scale bar: 10 μm. Magnifications of 40×. (**A**,**C**,**E**,**F**) Statistical analysis: Student’s *t*-test (*** *p* < 0.001).

**Figure 9 ijms-25-02814-f009:**
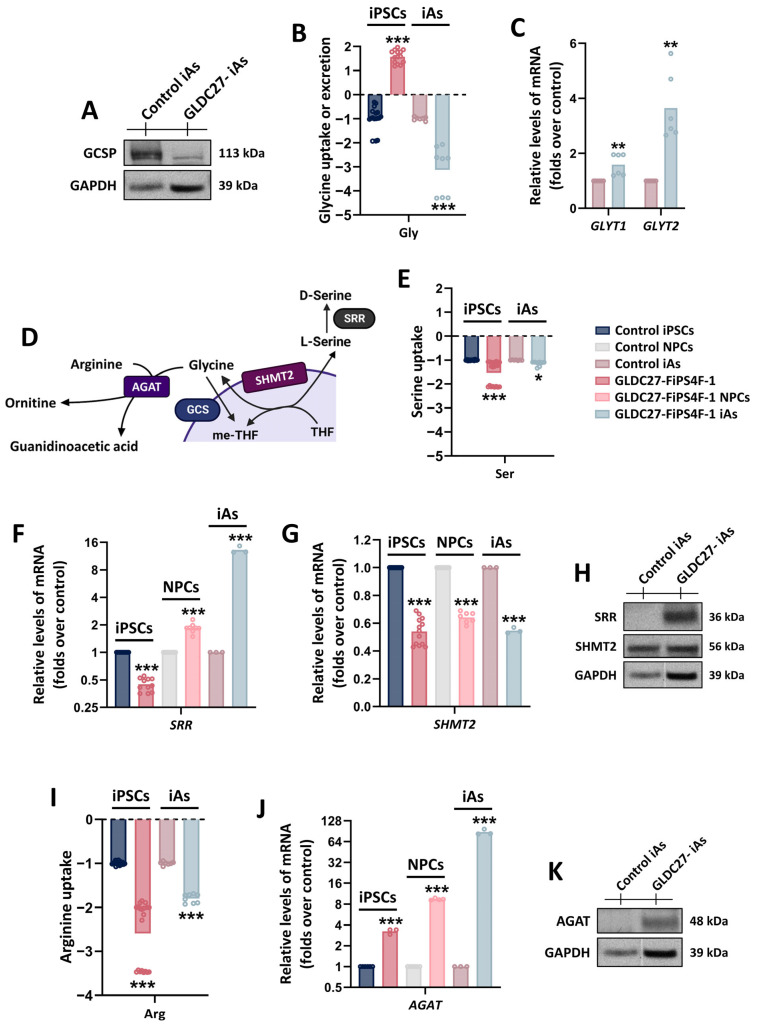
Glycine homeostasis in *GLDC*-deficient iA culture. (**A**) Representative Western blot showing GCSP protein levels in iA cultures. (**B**) Relative quantification of glycine extracellular levels in control and GLDC27-FiPS4F-1 iPSCs and iAs after 72 h of culture. (**C**) *GLYT1* and *GLYT2* gene expression levels (involved in glycine transport) in *GLDC*-deficient iAs compared to control iAs. (**D**) Schematic representation of SHMT2, SRR, and AGAT enzymes’ involvement in serine and creatine metabolism, which are closely related to glycine. (**E**) Relative quantification of serine’s extracellular levels in control and GLDC27-FiPS4F-1 iPSC lines and iA cultures after 72 h of culture. (**F**) Relative quantification of *SRR* and (**G**) *SHMT2* genes in *GLDC*-deficient iPSCs, NPCs, and iAs compared to control. (**H**) Representative Western blot of SHMT2 and SRR proteins. (**I**) Relative quantification of arginine extracellular levels in control and GLDC27-FiPS4F-1 iPSCs and iAs after 72 h of culture. (**J**) Relative quantification of *AGAT* gene in *GLDC*-deficient iPSCs, NPCs, and iAs compared to control line. (**K**) AGAT protein levels in iA cultures. (**A**,**H**,**K**) GAPDH was used as a loading control. “GLDC27-iAs” refers to GLDC27-FiPS4F-1 iA culture. (**B**,**E**,**I**) Control iPSCs and iAs were given a value of −1. The dotted line represents the 0 value. Positive levels represent release of the metabolite into the medium, while negative levels represent uptake from it. All values obtained in the measurement were normalized by protein. A total of *n* = 10 samples were evaluated. (**C,F**,**G**,**J**) Data were standardized against the endogenous *ACTB* gene and analyzed in triplicate. (**B**,**C**,**E**,**G**,**I**,**J**) Statistical analysis: Student’s *t*-test (* *p* < 0.05; ** *p* < 0.01; *** *p* < 0.001).

## Data Availability

Data contained within the article.

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
