# Peer review of "Metabolic Rewiring and Altered Glial Differentiation in an iPSC-Derived Astrocyte Model Derived from a Nonketotic Hyperglycinemia Patient"

_ijms, 2024, doi:10.3390/ijms25052814_

Round 1
Reviewer 1 Report
Comments and Suggestions for Authors
In the manuscript titled 'Metabolic Rewiring and Altered Glial Differentiation in an iPSC-derived Astrocytes Model from a Nonketotic Hyperglycinemia Patient,' the authors investigate the impact of GLDC gene variants on cellular fitness during the differentiation process of human induced pluripotent stem cells (iPSCs) into iPSC-derived astrocytes. They aim to identify sustainable mechanisms capable of overcoming GLDC deficiency. The following comments are provided to the Authors in a fully collaborative spirit, aiming to contribute to further improving this excellent study. The work is well-written, structured, and concise. The topic aligns perfectly with the journal, and the manuscript covers it in an objective and analytical manner.However, I have several minor concerns that the authors should address to strengthen the manuscript:
1. The authors should enhance the fluidity and clarity of the introduction to ensure better reader comprehension
2. In Figures 2A and 3A, the authors should consider separating the panels from the molecular pathway. The combination of these elements may contribute to confusion in the figure.
3. In panel 1D, could the authors change the scale to include both the control and NKH samples, as the very low values for NKH do not compare well on the graph?
Author Response
We thank the comments and recommendations of the reviewer for improving the final manuscript.
In this new version we have taken into consideration all the referee suggestions.
- The authors should enhance the fluidity and clarity of the introduction to ensure better reader comprehension.
Introduction has been rewritten to include relevant data missed in the previous version that could enhance the comprehension.
- In Figures 2A and 3A, the authors should consider separating the panels from the molecular pathway. The combination of these elements may contribute to confusion in the figure.
Following referee suggestions, Figures 2 and 3 have been reformatted to prevent confusion.
- In panel 1D, could the authors change the scale to include both the control and NKH samples, as the very low values for NKH do not compare well on the graph?
We have changed the scale in Panel 1D to highlight the differences between the control and NKH samples. Another question is the GCS activity in fibroblasts, which is absent in both control and patient cells since GCSP protein does not express in these cells. In this case, we include this data as another indication of the relevant role of GCS activity in pluripotent cells, reinforced by the data of reprogramming from somatic fibroblasts to iPSCs, which ends with the activation of the complex.
Conclusions has been also reformulated.
Reviewer 2 Report
Comments and Suggestions for Authors
Research manuscript entitled, “Metabolic Rewiring and Altered Glial Differentiation in an iPSC-derived astrocytes model derived from a Nonketotic Hyperglycinemia patient” related to a rare neurological metabolic disorder NKH which is responsible for malformations as well as life-threatening neurological manifestations. This is an interesting work establishing the involved mechanism in the progression of neurometabolic disease. The work has been clearly described and correlated with the obtained outcomes to hypothesize the involved mechanism. Still I’m having some suggestions to improve the readership of the manuscript:
1. More information about iPSCs model should be provided in the manuscript.
2. Conclusion should be reframed to include all the outcomes in support of proposed hypothesis.
3. Figures quality should be improved.
Author Response
We thank the comments and recommendations of the reviewer for improving the manuscript.
In this new version we have taken into consideration all the referee suggestions.
- Introduction has been rewritten to include relevant data missed in the previous version that could enhance the comprehension.
- We have included data relative to the generation of the IPSc model, and biallelic changes in GLDC beard for the patient cells.
- Conclusion has been reframed.
- To improve the clarity of the results, we have changed the scale in Panel 1D to highlight the differences between the control and NKH samples. Figures 2 and 3 have been reformatted to prevent confusion.